# Sensory Motor Function Disturbances in Mice Prenatally Exposed to Low Dose of Ethanol: A Neurobehavioral Study in Postnatal and Adult Stages

Kamal Smimih [1,2], Bilal El-Mansoury [2], Fatima Ez-Zahraa Saad [2], Manal Khanouchi [2], Souad El Amine [2], Abdelmohcine Aimrane [2], Nadia Zouhairi [1], Abdessalam Ferssiwi [2], Abdelali Bitar [2], Mohamed Merzouki [1,†] and Omar El Hiba [2,*,†]

1. Biological Engineering Laboratory, Faculty of Sciences and Techniques (FST), Sultan Moulay Slimane University, Beni Mellal 23000, Morocco
2. Laboratory of Anthropogenic, Biotechnology and Health, Nutritional Physiopathologies, Neuroscience and Toxicology Team, Faculty of Sciences, Chouaib Doukkali University, El Jadida 24000, Morocco
* Correspondence: elhiba.o@ucd.ac.ma
† These authors contributed equally to this work.

**Abstract:** Prenatal alcohol exposure (PAE) refers to fetal exposure to alcohol during pregnancy through placental barrier transfer from maternal blood. The postnatal outcomes of PAE differ among exposed individuals and range from overt (serious) alcohol-related behavioral and neurophysiological impairments to covert (silenced) symptoms. The aims of the present investigation were to assess the postnatal neurobehavioral disturbances, particularly, motor coordination and sensory-motor function in mice with PAE. Female mice with positive vaginal plugs were divided into three groups: group 1: Et + Pyr: received two i.p injections of ethanol (1 g/kg) followed by pyrazole (100 mg/kg). Group 2: Pyr: received an i.p injection of pyrazole (100 mg/kg). Group 3: C: of saline controls received, in equal volume, saline solution (NaCl 0.9%). After birth, mice pups were weighed and subjected to behavioral tests for motor function screening using the motor ambulation test, cliff aversion, surface righting, and negative geotaxis, while at the adult stage, mice were subjected to the open field, rotarod, parallel bars, and static rods tests. Our data show an obvious decrement of body weight from the first post-natal day (P1) and continues over the adult stage. This was accompanied by an obvious impaired sensory-motor function which was maintained even at the adult stage with alteration of the locomotor and coordination abilities. The current data demonstrate the powerful neurotoxic effect of prenatal ethanol exposure on the sensory-motor and coordination functions, leading to suppose possible structural and/or functional neuronal disturbances, particularly the locomotor network.

**Keywords:** prenatal alcohol exposure (PAE); motor activity; motor coordination; behavioral tests; pyrazole; mice

## 1. Introduction

Fetal alcohol syndrome disorder (FASD) is a medical term referring to a wide range of pathological conditions arising from alcohol exposure, particularly during the gestational stage. Previous studies have reported that FASD affects up to 0.77% of the global population [1,2] and constitutes a heavy burden on the health care systems, with patients suffering from lifelong physical and cognitive disability, minor craniofacial anomalies, retardation of growth, along with numerous neurological complications leading to variable cognitive and behavioral deficiencies [3], which may negatively impact the socio-professional patient's life, leading: to diminished productivity, and even unemployment and homelessness [4].

The consumption of alcohol, in spite of being healthy at low doses, is highly recommended to be avoided in several pathological complications [5,6]. Moreover, maternal

alcohol consumption during pregnancy is considered one of the most harmful and poisonous exposures modes.

It is highly established that the developing central nervous system (CNS) is particularly vulnerable to the toxic effects of alcohol [7,8], which is highly damaging for the fetus as it might induce pre- and postnatal growth deficiencies, along with skeletal and craniofacial abnormalities and severe CNS alterations.

Prenatal exposure to alcohol is believed to deteriorate motor development and subsequently motor functioning [9]. Indeed, previous clinical investigations on children with FASD have reported the presence of severe motor skills abnormalities in both gross and fine motor functions, together with motor coordination imbalance [10–12]. In addition, daily maternal alcohol abuse during pregnancy has tremendous effects on motor function in later childhood [13–16]. Interestingly, deficits in motor circuits development is also reported in prenatally alcohol exposed children who do not meet the full criteria for FASD [10].

Otherwise, ethanol has a potent teratogen effect, inducing the genesis of apoptosis in the developing brain [17]. Indeed, among FASD brain abnormalities, microencephaly and neuronal loss are highly sustained. While, MRI studies have revealed, in children and adolescent prenatally exposed to ethanol, the presence of cerebellum atrophy along with a decrease of the vermis size [18,19]. Such finding was sustained as well in several animal models of developmental alcohol exposure, wherein a loss of cerebellar Purkinje cells and granule cells was documented [20,21].

Numerous behavioral tests are now developed and well-established to assess motor function and overall behavioral, and cognitive phenotypes of FASD [22,23], and have effectively imitated some FASD characteristics phenotypes through prenatal or neonatal ethanol treatment [24–28]. However, there are still some inconstancies regarding the effect of maternal alcohol exposure on the motor function in newborn rodents. This could be related to differences in ethanol dosage, timing of treatment, and time of testing, as well as the genetic susceptibility of the model to ethanol's effects [29]. Hence, the aim of the present study is to assess the effect of alcohol administration in mice prenatally exposed to ethanol (during brain development stage) on motor function and motor coordination in two different stages; the postnatal and adult to assess the reversibility/irreversibility of these neurobehavioral patterns in the two life stages by appropriate neurobehavioral tests.

## 2. Materials and Methods

### 2.1. Chemicals

Ethyl alcohol (Darmstadt, Germany), and pyrazole (purchased from Sigma-Aldrich (St. Louis. MO 63103 USA) were used as chemical agents.

### 2.2. Animals

The study was carried out completely on Albino mice obtained from the central animal-care facilities of the Faculty of Sciences, Chouaib Doukali University, El Jadida, Morocco. Mice were housed at constant room temperature (25 °C) on a 12-h dark–light cycle with free access to water and food during several generations in Plexiglas cages. Animals were treated in compliance with the Moroccan Ethic Committee of the Moroccan Society for Ethics and Animal Research (MECAR-MoSEAR, Ref. UCD-FS-01/2023, 1 February 2023), while all efforts were made to minimize the number of mice used as well as the suffering by the use of anesthetics and sacrifice of animals with severe body weight loss.

### 2.3. Treatments

A total of 18 virgin female mice aged 15 weeks were used for our experiments. Female mice were caged overnight with a male mouse and the day of appearance of the vaginal plug is considered as the first day (D0) of pregnancy. Then, female mice (28 g ± 4) with positive vaginal plugs were divided into 3 groups:Group 1: Et + Pyr: pregnant female mice (*n* = 6) received 2 i.p ethanol injections (1 g/kg) scheduled as follow: the first injection at D10 while the second is at D13. Sixtymin before ethanol injection, each mouse received an i.p

injection of pyrazole (100 mg/kg) (dissolved in physiological saline) to prevent the hepatic ethanol oxidation leading to liver injury, by inhibiting the alcohol dehydrogenase [30]. All used solutions were sterilized 10 min prior each use, while the administrations were carried out between 10 to 12 AM. Group 2: Pyr: Pregnant female mice (*n* = 6) were injected only with pyrazole (100 mg/kg) (dissolved in physiological saline) at D10 and D13. Group 3: C: control mice (*n* = 6) received an equal volume of physiological saline solution (0.9% NaCl) (Figure 1).

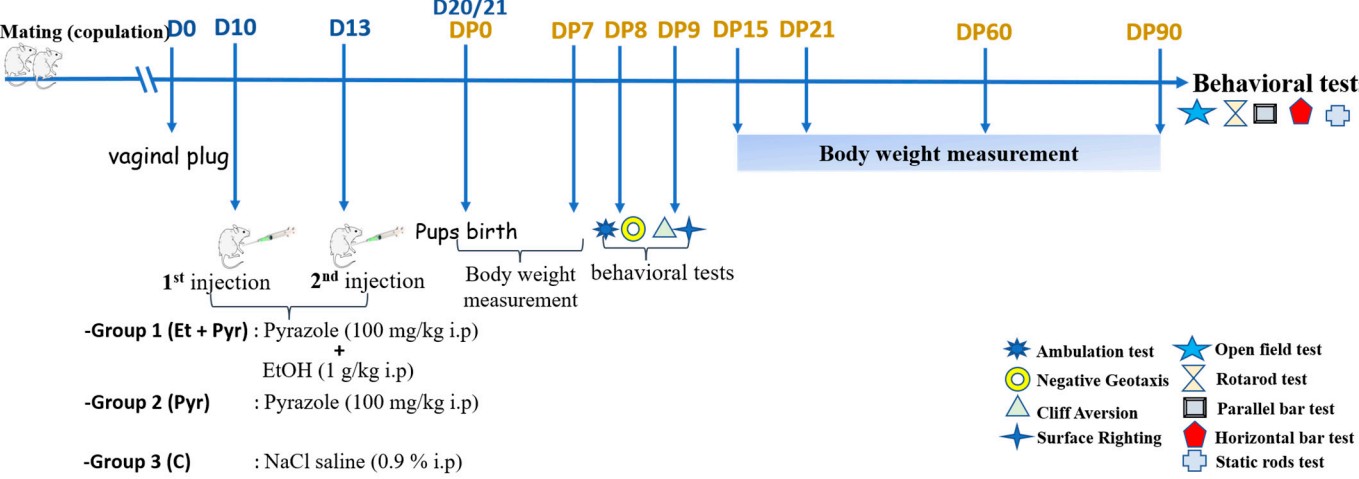

**Figure 1.** Experimental design and timeline schedule of the experiments.

*2.4. Morphometric and Behavioral Tests*

2.4.1. Body Weight Measurement

The evolution of the body weight of the offspring mice was monitored in the three groups through the three postnatal months at the P0, P7, P15, P21, P60, and P90.

2.4.2. Body Length Measurement

At the postnatal P14 in adult mice (and(weeks), the body length was manually measured from the tip of the nose to the base of the tail in the control and ethanol intoxicated mice.

2.4.3. Behavioral Tests

Ambulation Test

This test was applied to assess the motor function (activity and coordination) which was carried out at the 8th day of birth (P8). The test was performed according to Schussler and Ferguson, 2016. Each mouse was placed individually on a transparent plate (50 cm × 40 cm) with a moderate luminosity (100 lx) so as to be visible. Each animal behavior was studied for 3 min. A score of zero was given if there was no movement, a score of 1 if the animal cramps with asymmetrical movements, a score of 2 when the animal cramps with symmetry with a slow rhythm, and a score of 3 if the mouse cramps quickly with symmetrical steps or walks [31].

Negative Geotaxis

This behavioral test was used to assess motor coordination in young mice. The test was performed on the 9th day after the mice births. Normally, when mice are placed on their four legs on an oblique plane, they naturally turn on their side which is an innate behavior. This behaviour develops in mice between 3 and 15 days after birth [31]. Each mouse was placed facing down a slope (45° inclined plane) and blocked for 5 s, then released and the time spent to turn 180° was noted [31].

Cliff Aversion

The cliff aversion test was used to evaluate motor coordination [32]. This test relies on the innate fear of the mouse to turn away from a steep cliff. Each mouse (pups) was placed on the edge of a cliff (a table) with its legs and head on the edge. Fall avoidance is manifested by turning away from the edge of the device in the opposite direction [31].

Surface Righting

The surface righting reflex was also used to evaluate motor function, based on the ability of a mouse to turn from the dorsal to ventral position. This behavioral test was carried out on the 8th postnatal day. Each mouse was placed on its back on a cotton sheet and kept in that position for 5 s, then the mouse was released and the time it spends to return to its prone position was recorded [31].

Open Field Test

The open field (OF) test was also used to assess motor activity in mice. OF is a square enclosure apparatus with a surface area of $0.5m^2$ with a height of 40 cm (50 cm $\times$ 50 cm $\times$ 40 cm), and the bottom surface is subdivided into 25 identical squares. All mice were habituated to the OF two days before the test [33,34]. On the test day, each mouse was placed in the central area of the OF. Each animal's behavior was filmed for 5 min by a camera placed at the top of the apparatus. The videos were then examined and the number of crossed squares for each mouse running through was counted for each group; (the square is considered to be covered when the animal crosses it with all its legs) and used as an indication of motor activity [35].

Rotarod Test

The Rotarod test was used to assess motor coordination in mice and their ability to balance on a 30 mm diameter rotating rod [34]. Each mouse was placed on the rotating rod (speed of 12 rpm) away from the direction of rotation and 5 min was given to each animal as a maximum time of latency to fall. The behavior of each animal was recorded [36].

Parallel Bar Test

This test is suitable for the study of motor coordination in mice. The apparatus is composed of two metal bars, each 1 meter (1 m) long and 4 mm in diameter, and they were placed on a wooden platform raised 60 cm above a wooden support. The metal bars were equidistant from each other (30 mm apart) along the entire length of the apparatus, and a stopwatch was used to record the time (in seconds) spent for each animal to turn a 90° on the double bar, and the time spent by each mouse to travel to one end of the bars. Each animal behavior was given a particular score as described in the original protocol by Deacon [37].

Horizontal Bars Test

This test was also used to evaluate motor coordination in our mice. The apparatus consists of steel bars, 38 cm long, held 49 cm above the surface of the bench by a wooden support column at each end. The columns are attached to a stable wooden base. The diameters of these bars vary between 2, 4, and 6 mm, with the test starting from the narrowest 2 mm bar to the thickest bar. The procedure and the score obtained were conducted as previously described by [34,35].

Static Rods Test

Five wooden rods, each 60 cm long and of the following diameters (35, 28, 22, 15, 9 mm) were stapled to a shelf 60 cm above the ground. The end of the rod near the shelf has a mark at 10 cm to indicate the finish line. The mouse was placed at the free end of the wider rod and two measurements were taken: orientation time (time taken to orientate

180° from the starting position to the shelf) and transit time for each animal (time taken to reach the end of the shelf) [37].

Statistical Analysis

Statistical analysis was performed using SigmaPlot Software. Results were expressed as mean ± standard error mean (SEM). All data were analyzed by one-way ANOVA test followed by a post-hoc Tukey test. Values with $p < 0.05$ were considered as statistically significant.

## 3. Results

### 3.1. Body Weight Measurement

Our results showed a highly significant decrease of body wight in the ethanol + pyrazole-treated group (Et + Pyr) compared to control (C) and pyrazole groups (Pyr) from P0 to P90. However, there was no difference between the control group and pyrazole-treated mice (Figure 2).

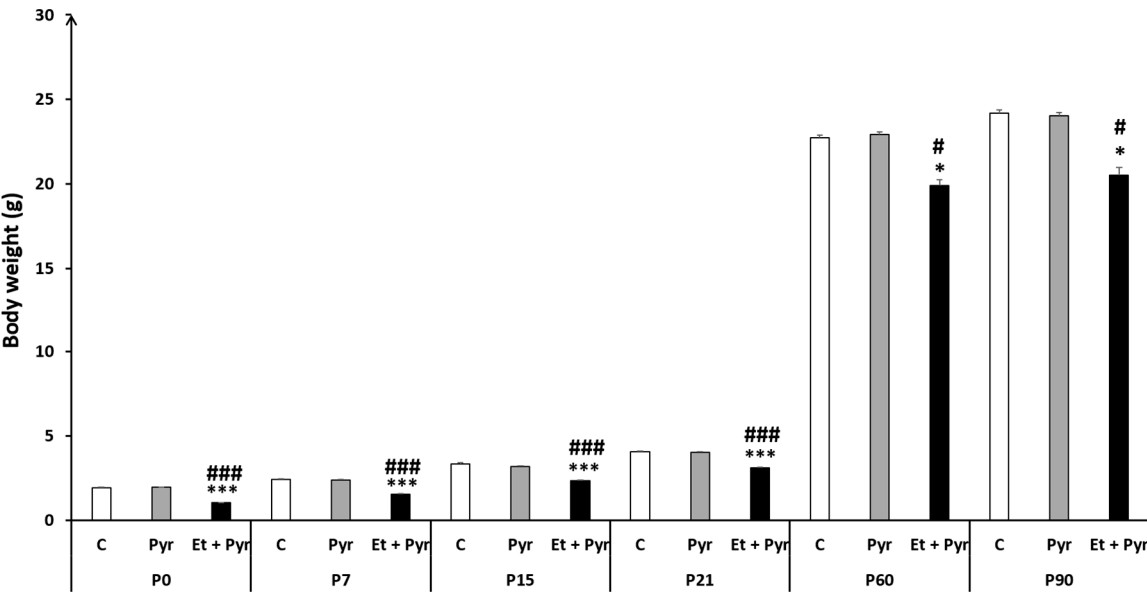

**Figure 2.** Histograms showing the average animal's body weight. C: control (*n* = 8), Pyr: pyrazole-treated mice (*n* = 8), Et + Pyr: ethanol + pyrazole-treated group (*n* = 8). Data are shown as group mean values ± S.E.M. \* $p < 0.05$, \*\*\* $p < 0.001$ vs. C, # $p < 0.05$, ### $p < 0.001$ vs. Pyr.

### 3.2. Body Length Measurement

The measurement of the body length shows a highly significant decrement of body length in the ethanol + pyrazole-treated group (Et + Pyr) compared to the control ($p < 0.001$) group at the age of P15. Such deficiency appears to be mainlined even at the adult age (15 weeks) ($p < 0.05$) (Figure 3).

### 3.3. Neurobehavioral Study

3.3.1. Ambulation Test

Our results showed a highly significant decrease of the ambulation score ($p = 0.032$) in the ethanol + pyrazole-treated group compared to the control (Figure 4), however, there was no difference between the control and pyrazole-treated groups ($p = 0.207$) as well as between the pyrazole group and treated group (0.151).

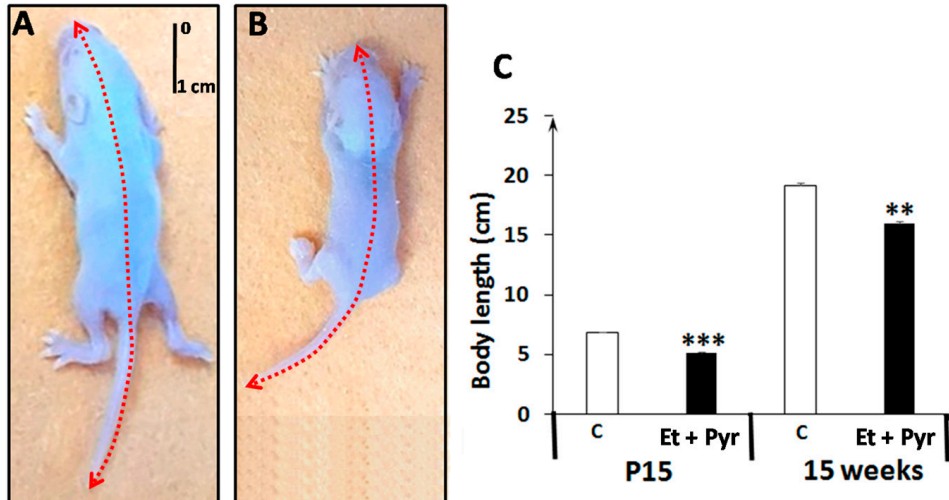

**Figure 3.** Photography of mice pups at the P15 age from control (**A**) and ethanol + pyrazole group (**B**). (**C**): graphical representation showing the average of animals' body length. Data are shown as group mean values ± S.E.M. ** $p < 0.05$ vs. C, *** $p < 0.001$ vs. C.

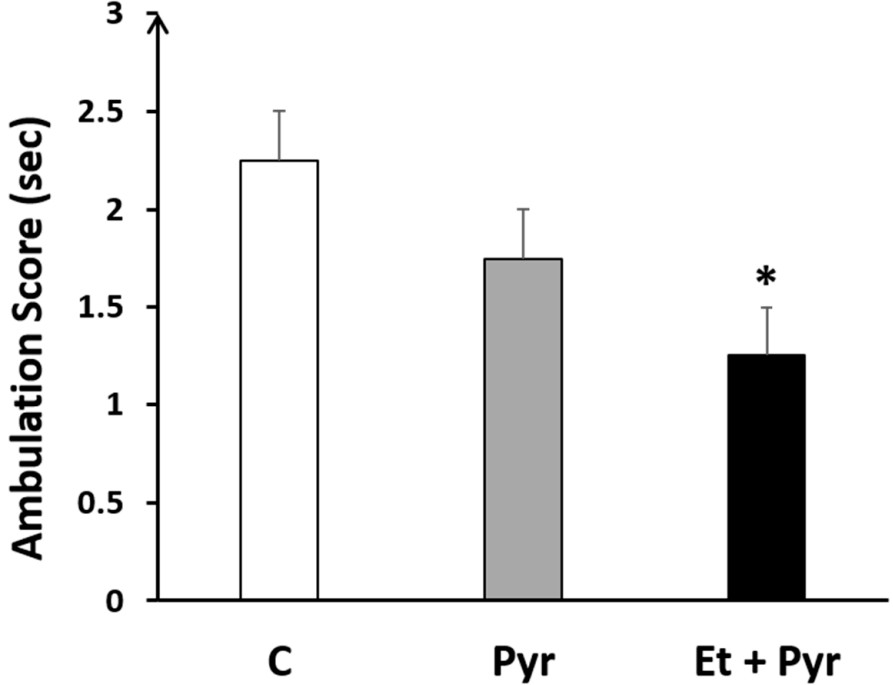

**Figure 4.** Histogram showing the ambulation score in the ethanol + pyrazol (Et + Pyr) (*n* = 8), the pyrazole (Pyr) (*n* = 8) and the control (C) groups (*n* = 8). Data are shown as group mean values ± S.E.M. * $p < 0.05$ vs. C.

3.3.2. Negative Geotaxis Test

Our results show a highly increased time spent to turn in the negative geotaxis task in the ethanol + pyrazole-treated group compared to the control ($p < 0.001$) and pyrazole ($p < 0.001$) groups (Figure 5). However, there was no difference between pyrazole and control groups ($p = 0.620$).

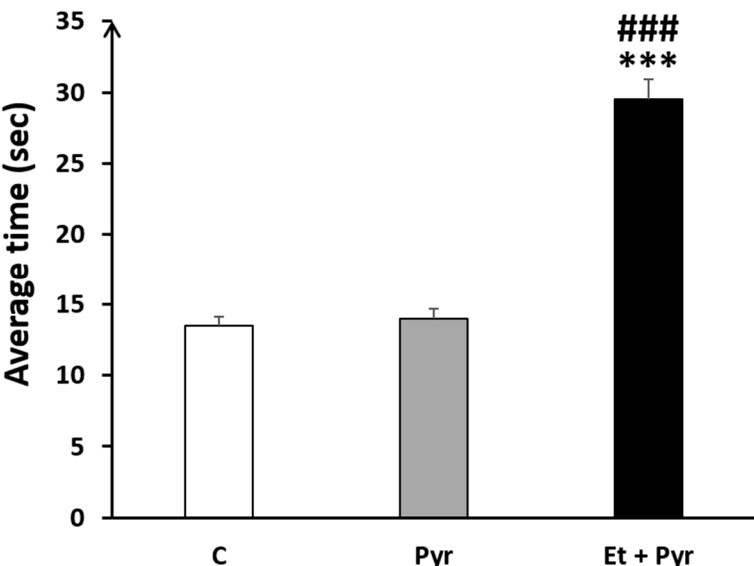

**Figure 5.** Graphical representation showing the average time spent to turn up in negative geotaxis task. C: control (*n* = 8), Pyr: pyrazole-treated mice (*n* = 8), Et + Pyr: ethanol + pyrazole-treated group (*n* = 8). Data are shown as group mean values ± S.E.M. *** $p < 0.001$ Et + Pyr vs. C. ### $p < 0.001$ Et + Pyr vs. Pyr.

### 3.3.3. Fall Avoidance Test

Our results show a slight decrease of fall avoidance percentage in the ethanol + pyrazole-treated group compared to the control and pyrazole groups (Figure 6), but still not significant statistically.

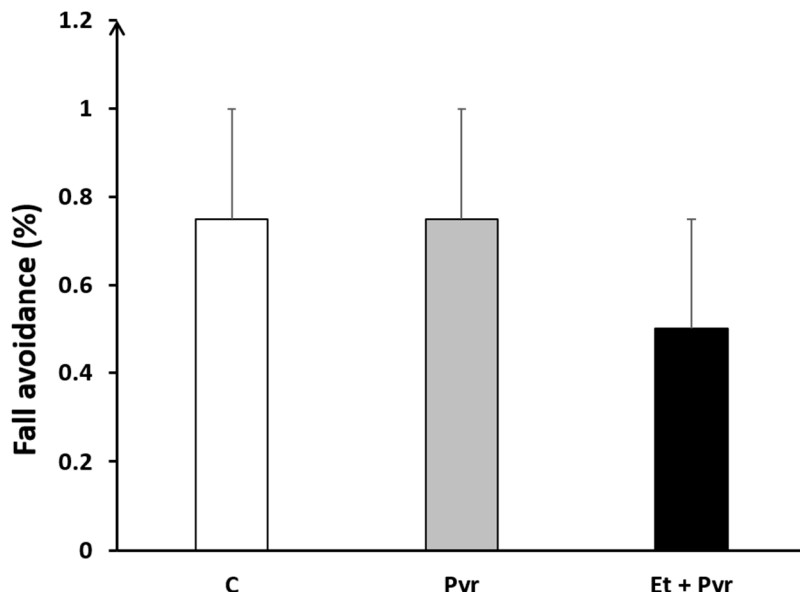

**Figure 6.** Graphical representation showing fall avoidance percentage. C: control (*n* = 8), Pyr: pyrazole-treated mice (*n* = 8), Et + Pyr: ethanol + pyrazole-treated group (*n* = 8). Data are shown as group mean values ± S.E.M.

### 3.3.4. Rears Test

Our results show the absence of any significant difference regarding the number of rears among the three studied groups; C, Pyr, and Et + Pyr (Figure 7).

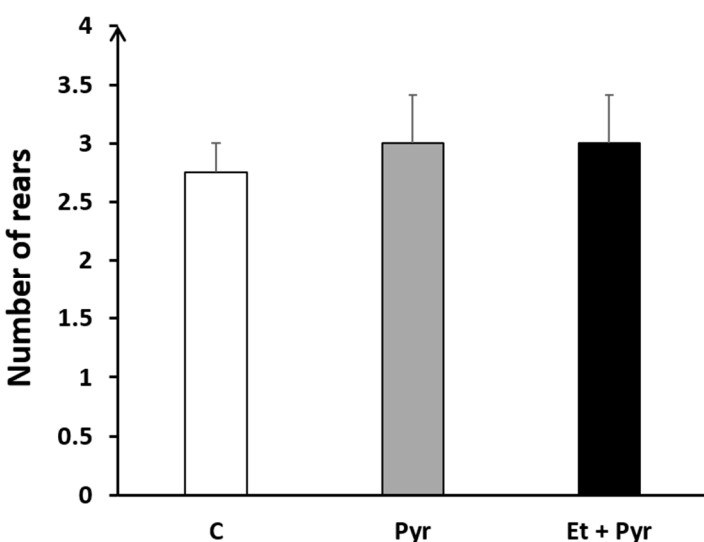

**Figure 7.** Histogram showing the number of rears. C: control (*n* = 8), Pyr: pyrazole-treated mice (*n* = 8), Et + Pyr: ethanol + pyrazole-treated group (*n* = 8). Data are shown as group mean values ± S.E.M.

### 3.3.5. Static Rods Test

✓ **Average time to turn 180°**

Our results show a highly significant time spent to turn 180° in the different rod sizes, respectively (Figure 8A–C); A: 35 mm, B: 28 mm, C: 22 mm in the ethanol + pyrazole-treated group as compared to the control (*p* < 0.001, *p* = 0.002, *p* = 0.001) and pyrazole (*p* = 0.004, *p* = 0.008, *p* = 0.004) groups.

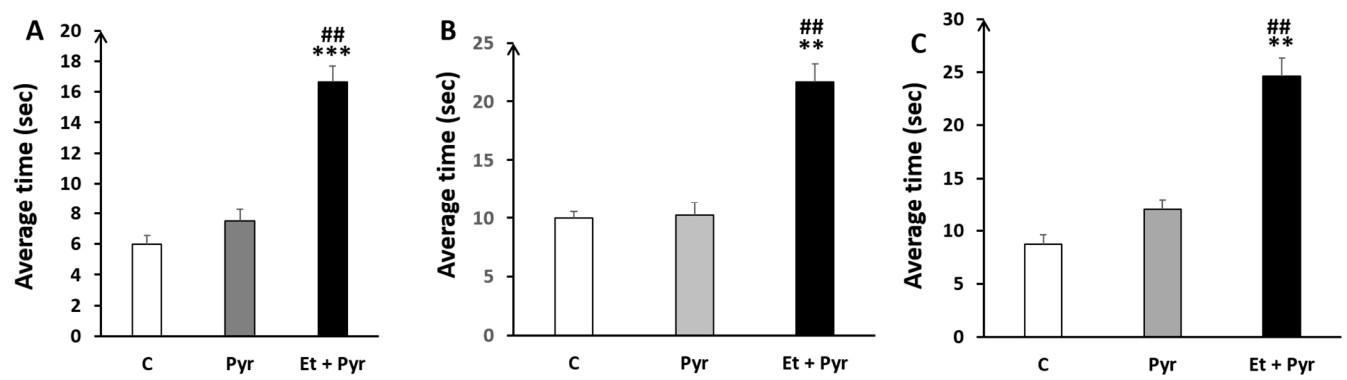

**Figure 8.** Histograms showing the average time to turn 180° on the static rods. C: control (*n* = 8), Pyr: pyrazole-treated mice (*n* = 8), Et + Pyr: ethanol + pyrazole-treated group (*n* = 8). (**A**): Rod (35 mm), (**B**): Rod (28 mm), (**C**): Rod (22 mm). Data are shown as group mean values ± S.E.M.** *p* < 0.01, *** *p* < 0.001 vs. C, ## *p* < 0.01 vs. Pyr.

✓ **Time to reach the end of the rods**

Our results show a highly significant time spent to reach the end of the different rod sizes, respectively (Figure 9A–C); A: 35 mm, B: 28 mm, C: 22 mm in the ethanol + pyrazole-treated group as compared to the control (*p* ≤ 0.001, *p* ≤ 0.001, *p* ≤ 0.001) and pyrazole groups (*p* ≤ 0.00, *p* ≤ 0.001, *p* ≤ 0.001).

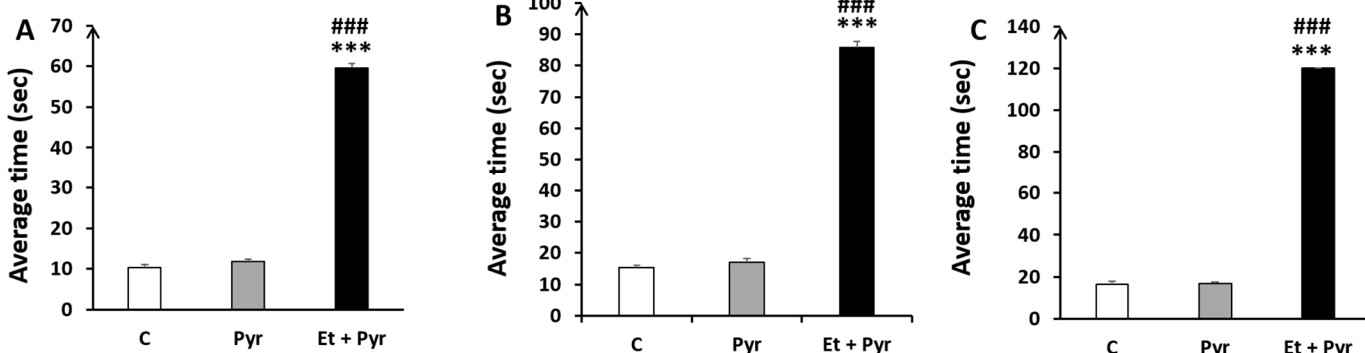

**Figure 9.** Histograms showing the average time to reach the end of the static rods. C: control (*n* = 8), Pyr: pyrazole-treated mice (*n* = 8), Et + Pyr: ethanol + prazole-treated group (*n* = 8). (**A**): Rod (35 mm), (**B**): Rod (28 mm), (**C**): Rod (22 mm). Data are shown as group mean values ± S.E.M. *** $p < 0.001$ vs. C, ### $p < 0.001$ vs. Pyr.

### 3.3.6. Parallel Bars

Our data show a highly significant increase of time spent to turn 90° on the parallel bars (Figure 10A) and time spent to travel at the end of the bars (Figure 10B) in the ethanol + pyrazole-treated group compared to the control and pyrazole groups.

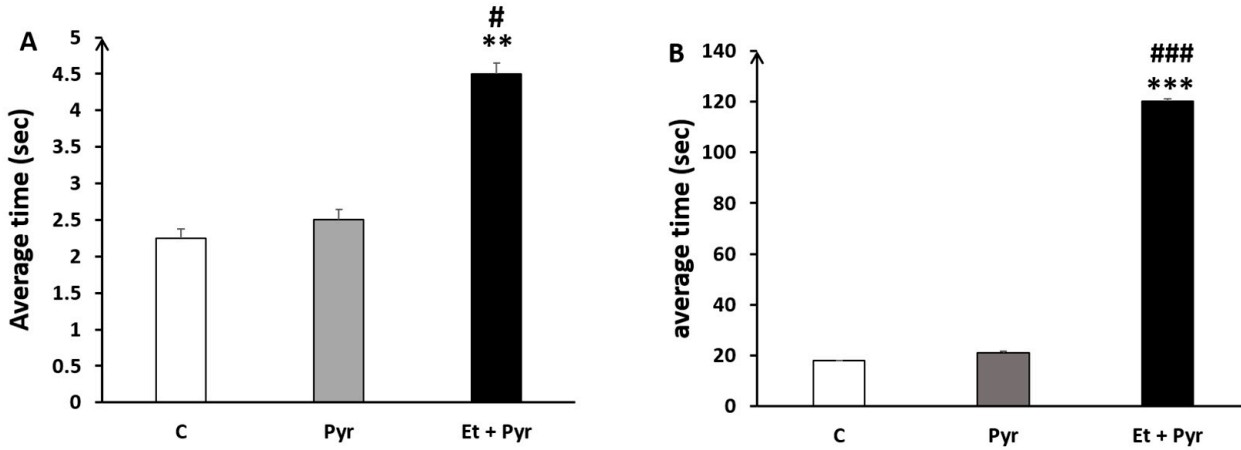

**Figure 10.** Histograms showing the average time spent to turn (**A**) and to reach the end of the parallel bars (**B**). C: control (*n* = 8), Pyr: pyrazole-treated mice (*n* = 8), Et + Pyr: ethanol + pyrazole-treated group (*n* = 8). Data are shown as group mean values ± S.E.M. ** $p < 0.01$, *** $p < 0.001$ vs. C, # $p < 0.05$, ### $p < 0.001$, vs. Pyr.

### 3.3.7. Horizontal Bars

This test was as well established to assess motor coordination in our mice. Our results showed a significantly decreased average score on the horizontal bars in the ethanol + pyrazole-treated group as compared to the control ($p = 0.018$) and pyrazole-treated mice ($p = 0.018$) (Figure 11).

### 3.3.8. Rotarod

Our results did not reveal any significant difference regarding the latency to fall among the three studied groups: C, Pyr, and Et + Pyr (Figure 12).

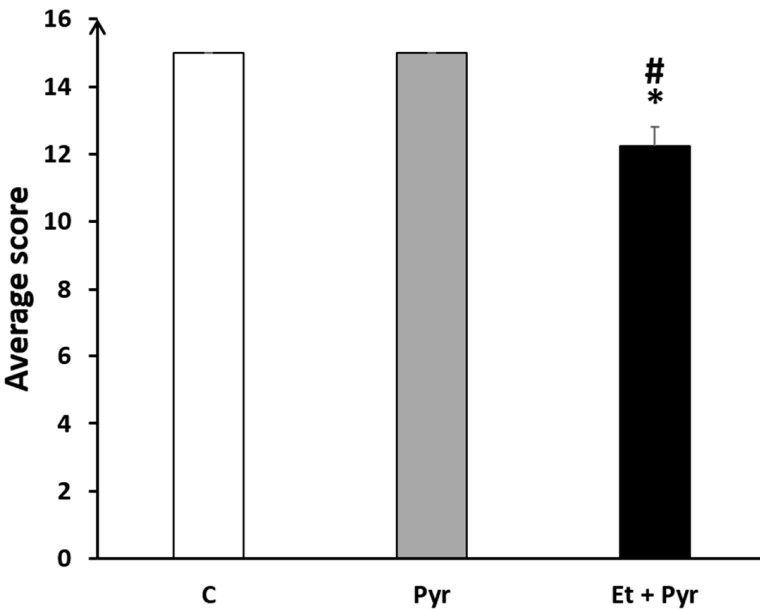

**Figure 11.** Graphical representation showing the average score in the horizontal bars. C: control ($n = 8$), Pyr: pyrazole-treated mice ($n = 8$), Et + Pyr: ethanol + pyrazole-treated group ($n = 8$). Data are shown as group mean values $\pm$ S.E.M. * $p < 0.05$ vs. C. # $p < 0.05$ vs. Pyr.

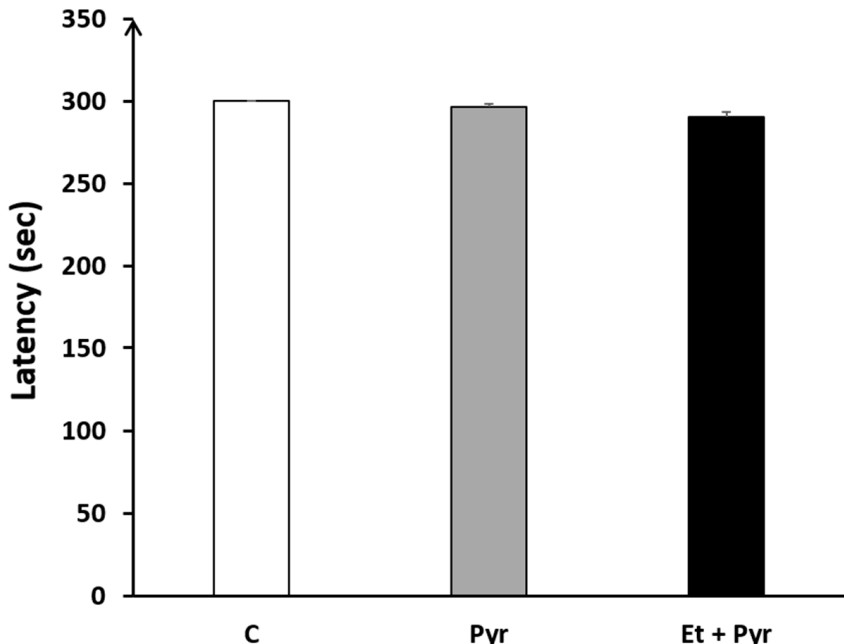

**Figure 12.** Graphical representation showing the latency to fall from the horizontal axe of rotarod. C: control ($n = 8$), Pyr: pyrazole-treated mice ($n = 8$), Et + Pyr: ethanol + pyrazole-treated group ($n = 8$). Data are shown as group mean values $\pm$ S.E.M.

### 3.3.9. Open Field Test

Using the open field test, our data showed a significant decrease of the number of crossed boxes in the ethanol + pyrazole-treated group compared to the control ($p = 0.004$) and pyrazole ($p = 0.007$) groups (Figure 13).

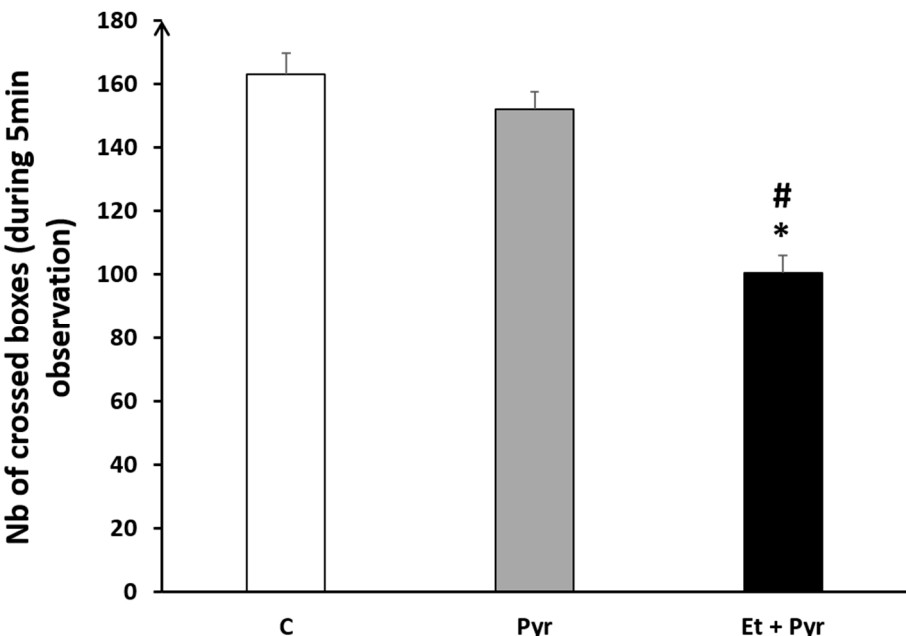

**Figure 13.** Graphical representation showing the number of crossed boxes in the open field (during 5 min observation). C: control (*n* = 8), Pyr: pyrazole-treated mice (*n* = 8), Et + Pyr: ethanol + pyrazole-treated group (*n* = 8). Data are shown as group mean values ± S.E.M. * $p < 0.05$ vs. C. # $p < 0.05$ vs. Pyr.

## 4. Discussion

Through the present study, we demonstrated an impairment of the sensory-motor function in mice intoxicated prenatally with a low dose of ethanol. Such disturbance is manifested by motor incoordination in newborn mice at the P8 and P9 stages; with an obvious abnormal reflex in the negative geotaxis and the ambulation tests which could be resulted from abnormal maturation of the brain structures controlling these functions. Strikingly, this impairment continues over the adult stage, wherein mice exhibit significant locomotor activity deficits in the open field, static bars, parallel bars, and horizontal bars tests. Additionally, this alteration was accompanied by a loss of the body weight starting from birth and maintained even at adulthood as well as an obvious decrease in body length, reflecting sever growth deficiency [38].

Indeed, a body of evidence seems to support the negative impact of alcohol exposure on the body weight in rodents and even in humans [39–41]. Thus, in pregnant mice orally intoxicated with 25% ethanol, some authors have reported decreased body weight at birth and adulthood stages [42]. However, previous reports appear to exclude any effect of prenatal ethanol exposure on body weight and growth. Indeed, Middaugh et al., using the C57BL/6cr mice model, reported that the body weight of the offspring mice fed with alcohol-based diets during pregnancy did not differ from that of controls until weaning [43]. Other studies reported no differences regarding litter size or pup weight in animal models of alcohol exposure during pregnancy [44–50]. These discrepancies among studies could be explained by the differences in the root of alcohol administration, the form and the dose used as well, as the animal model used.

Whereas, our alcohol-treated mice exhibited sensory-motor dysfunction, which may reflect deep impairment in the development of the brain motor networks. Similarly, in a study by Kleiber et al. (2011), performed in C57BL/6J mice, ethanol treatment induced by the two-bottle choice paradigm of maternal ethanol consumption throughout gestation and early postnatal period elicited motor deficits with decreased distance traveled and average speed movement in the open-field test, as well as a delay in neonatal reflex and coordination development [29]. Additionally, in a mice model of maternal binge-like alcohol drinking, authors showed an impairment of motor coordination in the late adolescent stage (PD48)

in the rotarod test [51]. However, there are still controversies about the effect of prenatal ethanol on locomotor activity, with some studies reporting no effect on motor functions. Indeed, in a mouse model of prenatal exposure to ethanol at a dose of 4.0 g/kg at the day 16 and 17 of pregnancy, mice did not exhibit any changes in locomotor activity and motor coordination in the open field test [52]. Whereas, Heck et al. reported no impairment regarding PAE at a lower dose (1.0 g/kg weight) in Swiss–Webster mice and rats, which may indicate a dose-dependent effect of PAE [53]. Such discrepancies among studies regarding motor function in PAE could be related to the model used, ethanol dose, and timing of administration [29].

Likewise, in humans, post-mortem studies have reported a wide range of abnormalities associated with FAS, including microcephaly, agenesis of the corpus callosum, and abnormal CNS development [51].

Although, motor impairment and motor incoordination could be the result from abnormal brain structure development induced by alcohol toxicity. In addition to behavioral alterations, histological and electrophysiological disturbances were reported in offspring of ethanol-drinking females [54]. It is highly established that upper cortical layer neurons are predominantly generated at embryonic days 16 and 17 in mice [52], and any lesion of brain development could have deleterious effects. Even though some brain regions involved in motor learning, such as the cerebellum, wherein neurogenesis occurs postnatally in mice [55], PAE could trigger motor abnormalities. Whereas, ethanol administered to C57B1/6J mice on gestational day 8 resulted in volume reduction in several regions of the brain, while the cerebellum was among the most affected regions [56]. In another study performed by Pierce et al., the authors have reported neuronal loss in the cerebellum, wherein Purkinje cells were the most reduced cells in addition to cerebellar granule cells, which were significantly reduced in the granular layer, but not in the external granular layer in an animal model of PAE [57]. More recently, Servais and coworkers studied cerebellar Purkinje cells in vivo and in vitro after prenatal ethanol exposure and they reported an increase in Purkinje cell firing and rapid oscillations of local field potential observed, which might underlie motor coordination abnormalities and motor impairment observed in FAS [54].

## 5. Conclusions

Through the present investigation, we emphasized the neonatal neurobehavioral outcomes of prenatal ethanol exposure in mice. In addition to the body weight loss, our mice exhibited severe and persistent sensory motor dysfunction, as well as motor incoordination during the early postnatal life which extends to the adulthood age. Such neuro-behavioral deficits may rise from a possible neuronal dysfunction, particularly the locomotor networks. However, the mechanism by which alcohol induces these abnormalities is still not fully understood, therefore, further studies are needed to well understand the underlying mechanisms, allowing focus on the appropriate therapeutic strategies by targeting the specific neuronal and/or glial compounds.

**Author Contributions:** Conceptualization: O.E.H.; methodology, O.E.H.; software: K.S., B.E.-M. and A.A.; validation, O.E.H. and M.M.; formal analysis, A.F. and A.B.; investigation, K.S.; resources; A.F. and A.B.; data curation, F.E.-Z.S., M.K., S.E.A.; writing—original draft preparation, K.S.; writing—review and editing, O.E.H.; visualization, N.Z.; supervision, O.E.H. and M.M.; project administration, O.E.H. and M.M.; funding acquisition, A.F. and A.B. All authors have read and agreed to the published version of the manuscript.

**Funding:** This research was funded by the FSJ-UCD for research units funding (annual budget).

**Institutional Review Board Statement:** The study was conducted in accordance with the Declaration of Helsinki, and approved by the Institutional Review Board of the Moroccan Ethic Committee of the Moroccan Society for Ethics and Animal Research (MECAR-MoSEAR, Ref. UCD-FS-01/2023, 1 February 2023).

**Data Availability Statement:** Not applicable.

**Conflicts of Interest:** The authors declare no conflict of interest.

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
