# Peer review of "Sensory Motor Function Disturbances in Mice Prenatally Exposed to Low Dose of Ethanol: A Neurobehavioral Study in Postnatal and Adult Stages"

_2035-8377, doi:10.3390/neurolint15020036_

Round 1

Reviewer 1 Report

In the current manuscript, Kamal S and co-workers investigated the sensory-motor function disturbances in mice prenatally exposed to a low dosage of ethanol. Despite previous studies demonstrating various behavioral outcomes in mice given low dosages of ethanol during the gestation period, the current study highlights that along with sensory motor disturbance, prenatal ethanol intoxication was accompanied by a change in body weight starting from birth that was maintained even at adulthood.

Overall, the study seems interesting with a few modifications that might improve the quality of the paper.

1) Throughout the manuscript there are various spelling mistakes and     

     grammatical errors example:

line no # 40 hugly

line no # 43 avoidedin           

line no # 43 Morover

line no# 81 (Darmstadt,Germany)

This issue disturbs the readership. The authors may need a though review before it gets a publication-quality manuscript.       

2) Although the authors described the number of pregnant female mice taken in the study however through the manuscript the sample size was missing for each behavioral test. It would be better if the author describes current datasets in scatter dot plots for histograms. Also clearly mention the animal number in the figure legends.

3) The significant finding of the current study is the reduction in body weight as described in figure 1. It would be better if the author could provide individual groups' body lengths. This would help to rule out any metabolic issues with pups or at the latter stage.  

4)  Given that mice exhibit sensory motor behavior deficit following ethanol exposure and the author proposes that there might be neuronal structural and/or functional disturbance. Like a previous finding by Abbott, et al., 2017 authors should provide the brain images of each individual group in support of this argument.

5)  Please also provide the relevant datasets to support your finding, whether the reduction in body weight is due to a decrease in feeding or anhedonia caused by prenatal ethanol exposure.

Author Response

Reviewer 1

In the current manuscript, Kamal S and co-workers investigated the sensory-motor function disturbances in mice prenatally exposed to a low dosage of ethanol. Despite previous studies demonstrating various behavioral outcomes in mice given low dosages of ethanol during the gestation period, the current study highlights that along with sensory motor disturbance, prenatal ethanol intoxication was accompanied by a change in body weight starting from birth that was maintained even at adulthood.

Overall, the study seems interesting with a few modifications that might improve the quality of the paper.

1) Throughout the manuscript there are various spelling mistakes and     

     grammatical errors example:

line no # 40 hugly

line no # 43 avoidedin           

line no # 43 Morover

line no# 81 (Darmstadt,Germany)

This issue disturbs the readership. The authors may need a though review before it gets a publication-quality manuscript. 

Thank you for your time and your valuable comments. Indeed, the whole manuscript was carefully revised again to correct all these mistakes and the English style as well was revised by native English speaker.

2) Although the authors described the number of pregnant female mice taken in the study however through the manuscript the sample size was missing for each behavioral test. It would be better if the author describes current datasets in scatter dot plots for histograms. Also clearly mention the animal number in the figure legends.

Thank you for your pertinent remark, indeed, we revised all the figures legends and the number of animals used was included

3) The significant finding of the current study is the reduction in body weight as described in figure 1. It would be better if the author could provide individual groups' body lengths. This would help to rule out any metabolic issues with pups or at the latter stage.  

Thank you for your suggestion, indeed, we have included new figure with histogram showing the body length in the ethanol group and control (fig 3)

4)  Given that mice exhibit sensory motor behavior deficit following ethanol exposure and the author proposes that there might be neuronal structural and/or functional disturbance. Like a previous finding by Abbott, et al., 2017 authors should provide the brain images of each individual group in support of this argument.

Indeed, the same study we are performing will also focus on the histological changes in some brain areas involved in the control of motor function and that from the prenatal to the adult stages to show the kinetic evolution of these cellular/histological abnormalities underlying the motor coordination and motor function deficiencies. The study will be completed in some few months to be completed and will be subjected to a future publication in the same journal

5)  Please also provide the relevant datasets to support your finding, whether the reduction in body weight is due to a decrease in feeding or anhedonia caused by prenatal ethanol exposure.

Concerning feeding, all groups were given the same amount of barely during all the experiment duration, and there were no signs of anhedonia as the feeding behavior was unaffected. This could be explained by the fact that alcohol crosses the placenta barrier and eventually into the bloodstream of the fetus might be responsible for the body weight decrement in our ethanol-pyrazole treated group as PAE known to cause metabolic dysregulation in the offspring.  

Reviewer 2 Report

The manuscript by Smimih et al., investigates the low doses of ethanol toxicity in prenatal alcohol exposure (PAE), assessing several neurobehavioral tests. They discuss persistent sensory-motor dysfunction as well as motor incoordination during the early postnatal life and adulthood stages.

The experimental plan is sufficiently executed, but the data are not clear and well-written. The methods description is correct, and obviously, in the discussion are evidenced many limits of these descriptive results.

The authors reported that ‘A total of 18 virgin female mice aged of 15 weeks were used for our experiments’. Has the minimum number of replications to be used in an experiment been calculated? The minimum sample size can be determined with a power analysis. This must be done prior to carrying out the experiment. It is useful for determining the number of animals needed to achieve an achievement statistically significant, avoiding sacrificing animals unnecessarily (neither too many nor too few).

This is a limited report also due to the not sufficient writing and exposition. Several typos, misuse, and grammatical errors are present throughout the entire manuscript and an extensive revision is necessary. The literature cited is of important relevance, but this manuscript lacks novelty in the field of fetal alcohol syndrome.

Minor points:

- the employment of pyrazole was not explained;

-funding is not well indicated.

For the mentioned reasons, the manuscript is unacceptable as is, but worth reconsideration if extensively revised.

Author Response

Reviewer 2

The manuscript by Smimih et al., investigates the low doses of ethanol toxicity in prenatal alcohol exposure (PAE), assessing several neurobehavioral tests. They discuss persistent sensory-motor dysfunction as well as motor incoordination during the early postnatal life and adulthood stages.

The experimental plan is sufficiently executed, but the data are not clear and well-written. The methods description is correct, and obviously, in the discussion are evidenced many limits of these descriptive results.

The authors reported that ‘A total of 18 virgin female mice aged of 15 weeks were used for our experiments’. Has the minimum number of replications to be used in an experiment been calculated? The minimum sample size can be determined with a power analysis. This must be done prior to carrying out the experiment. It is useful for determining the number of animals needed to achieve an achievement statistically significant, avoiding sacrificing animals unnecessarily (neither too many nor too few).

We sincerely thank the reviewer for his insightful remarks; indeed, the number of mice used in the current experiment was calculated via the equation:

Corrected sample size = Sample size/ (1− [% attrition/100]) taking into consideration the mortality rate in females and intrauterine deaths.

This is a limited report also due to the not sufficient writing and exposition. Several typos, misuse, and grammatical errors are present throughout the entire manuscript and an extensive revision is necessary. The literature cited is of important relevance, but this manuscript lacks novelty in the field of fetal alcohol syndrome.

Indeed, we apologies the English mistakes, the whole manuscript was revised again by a native English speaker

Minor points:

  1. - the employment of pyrazole was not explained;

Thank you for your remark, indeed, we included a statement in material and methods section (3-3-treatment) explaining the utility of the use of pyrazole

  1. -funding is not well indicated.

The study among others in our laboratory was fully funded by the annual budget attributed by the university to each research team including our laboratory. This was stated in the funding section

For the mentioned reasons, the manuscript is unacceptable as is, but worth reconsideration if extensively revised.

Thank you so much for all your consistent remarks, we hope that the manuscurpt in its new format will meet your expectations.

Reviewer 3 Report

Sensory motor function disturbances in mice prenatally ex- 2 posed to low dose of ethanol: a neurobehavioral study in postnatal and adult stages

The study demonstrates the neurotoxic effect of ethanol administration during the prenatal stage. The approach and the overall design of the study are good. However, the authors should address some minor concerns.

1.       The article demonstrates the effect of “low dose ethanol”. Justify? The authors need to revise the introduction by citing the previous studies conducted using different doses of ethanol in prenatal stages.

2.       Abstract says “the powerful neurotoxic effect of a single dose of ethanol administrated during the prenatal stage “- However there are two intraperitoneal injections of ethanol

3.       How did the authors fix the dose of ethanol and pyrazole?

4.       Add the future implications of the study at the end of the discussion.

Author Response

Reviewer 3

The study demonstrates the neurotoxic effect of ethanol administration during the prenatal stage. The approach and the overall design of the study are good. However, the authors should address some minor concerns.

  1. The article demonstrates the effect of “low dose ethanol”. Justify? The authors need to revise the introduction by citing the previous studies conducted using different doses of ethanol in prenatal stages.

The whole introduction was revised and rewritten to emphasize the previous literature reports showing the neurotoxic effect of prenatal ethanol exposure in rodents as well as in humans

  1. Abstract says “the powerful neurotoxic effect of a single dose of ethanol administrated during the prenatal stage “- However there are two intraperitoneal injections of ethanol

Apologies for such error, actually, we used two i.p   injections instead of one; this was corrected in the abstract and the conclusion parts

  1. How did the authors fix the dose of ethanol and pyrazole?

For ethanol, we were based on some literature reports using in mice a dose of 2g/kg (doi.org/10.1152/japplphysiol.00751.2003 and   DOI: 10.1016/0890-6238(93)90234-x ), however, our preliminary trials have revealed high intrauterine mortality rate. Therefore we were limited to two injections of 1mg/kg which was the most appropriate for us.

For Pyrazole, we adopted the protocol published by Ukita et al., 1993 ( DOI: 10.1016/0890-6238(93)90234-x) using a dose of 100 mg/kg  prior ethanol administration

  1. Add the future implications of the study at the end of the discussion.

The conclusion of the manuscript was partially rewritten to emphasize the future implications of the current study

Round 2

Reviewer 2 Report

The revised version of the manuscript improved all previous comments.

The manuscript is acceptable in its present form.